# RNA-Seq Analysis of Testes from Mice Exposed to Neodymium Oxide

**DOI:** 10.3390/toxics11120952

**Published:** 2023-11-22

**Authors:** Shurui Wang, Ning Bu, Yudan Yun, Xuemin Shi, Suhua Wang, Yanrong Gao

**Affiliations:** 1Baotou Medical College, Baotou 014042, China; wangsurri@163.com (S.W.); 18169521879@163.com (Y.Y.); sxmyyhx@126.com (X.S.); 2Center for Global Health, The Key Laboratory of Modern Toxicology of Ministry of Education, School of Public Health, Nanjing Medical University, Nanjing 211166, China; 15661346722@163.com; 3Wulanchabu Medical College, Wulanchabu 012001, China

**Keywords:** neodymium oxide, RNA sequencing, lncRNA, circRNA, mRNA, reproductive toxicity

## Abstract

(1) Objective: Rare earth neodymium oxide (Nd_2_O_3_) is refined and used extensively around the world, and the occupational and environmental safety of rare piles of the earth has attracted considerable attention. Nd_2_O_3_ enters the human body through the respiratory system, reaches various organs through blood circulation, and accumulates to produce toxic effects. At present, little is known about the reproductive toxicity of Nd_2_O_3_. Non-coding RNAs participate in a variety of physiological activities and are very important for spermatogenesis. However, it is unknown whether they are involved in Nd_2_O_3_-induced reproductive toxicity. Therefore, we conducted a pathological analysis, sperm quality testing, and RNA-seq on the testicular tissue of mice exposed to Nd_2_O_3_ to find the key genes and regulatory pathways of male reproductive damage and explore the early biomarkers and mechanisms of reproductive damage caused by Nd_2_O_3_. (2) Methods: After exposure of mice to Nd_2_O_3_, we carried out a pathological analysis and RNA-seq analysis for miRNAs/lncRNAs/circRNAs/mRNAs on the testicular tissue of mice, and the total RNAs were used to investigate miRNA/lncRNA/circRNA/mRNA expression profiles by strand-specific RNA sequencing at the transcriptome level to help uncover RNA-related mechanisms in Nd_2_O_3_-induced toxicity. (3) Results: Nd_2_O_3_ damaged testis and sperm morphology, significantly decreased the number of sperm, and deformed the sperm head and tail. RNA-seq analysis showed that the expression level of mRNA/miRNA/circRNA/lncRNA in the testicular tissue of mice exposed to Nd_2_O_3_ is abnormal. Gene Ontology (GO) and Kyoto Encyclopedia of Genes and Genomes (KEGG) analysis demonstrated that the functional enrichment of differentially expressed genes (DEGs) and their target genes was closely related to the related pathway of spermatogenesis. Furthermore, some miRNAs/lncRNAs/circRNAs that were greatly upregulated or inducibly expressed, implying their potential value as candidate markers for Nd_2_O_3_-induced reproductive toxicity, help us to further investigate the mechanisms of key genes, key signaling pathways, and inter-gene regulation for Nd_2_O_3_-induced reproductive toxicity. (4) Conclusions: This study provides the first database of a Nd_2_O_3_-induced transcriptome. This information is useful for the development of biomarkers of Nd_2_O_3_-induced reproductive injury and promotes understanding of the reproductive toxicity mechanism of Nd_2_O_3_.

## 1. Introduction

Rare earth elements, a new kind of renewable resource, are extensively used in automobiles, nuclear energy, the military, and other fields [1,2]. Nevertheless, with the increasing processing and utilization of rare earth elements, their potential occupational and environmental health safety has received considerable attention. Neodymium oxide (Nd_2_O_3_), as an important raw material for electrolytic neodymium, mainly comes into the human body through the respiratory system in the form of fine particles or aerosol and then circulates to various tissues and organs through the blood. Therefore, long-term exposure to Nd_2_O_3_ in mining, production, and processing of Nd_2_O_3_ and in drugs and industrial products leads to biological accumulation in the body and causes toxic effects. Some epidemiological and animal experimental studies, including our previous studies, found that Nd_2_O_3_ dust can cause pneumonia and pulmonary fibrosis [3,4]. Recent studies have shown that rare earth cerium oxide particles (CeO_2_NPs) can increase the content of Ce element in mouse testis, cause pathological damage to testicular tissue, and reduce sperm viability and survival rate [5]. Other studies also show that rare earth lanthanum oxide particles (La_2_O_3_ NPs) penetrate the blood–testis barrier of mice and accumulate in testicular tissue to cause apoptosis of spermatogenic cells [6]. However, the underlying mechanisms of reproductive damage have not been explicitly elucidated.

About 15% of couples in the world suffer from infertility, of which about 20–30% of cases are caused by male factors [7]. Male fertility depends on a continuous process of spermatogenesis from spermatogonial stem cells to sperm, which requires the correct temporal and spatial expression of specific genes in germ cells. At present, due to a lack of clear candidate genes related to human germ cells and infertility, the diagnosis and potential causes of male infertility have not been fully understood. However, environmental and occupational harmful factors alter gene expression. Environmentally induced sperm quality decline and reproductive damage have gradually been discovered. A retrospective study showed that environmental pollutants such as atmospheric particulate matter 2.5 (PM2.5) and carbon monoxide (CO) would interfere with sperm production and destroy the male reproductive system [8]. Misexpression of insulin-like growth factor 2 (IgF2) and H19 genes and changes in epigenetic mechanisms such as DNA methylation in the offspring of male mice exposed to arsenic result in damage to the reproductive system and reduced sperm quality [9,10]. Recently, we detected the expression profile of non-coding RNA (ncRNA) in the serum of rare earth pneumoconiosis patients. Interestingly, we found that the functions of some ncRNAs that were greatly upregulated or inducibly expressed were concentrated in the reproductive system [3].

The above studies suggest that changes in genes and epigenetics may be involved in the toxic reproductive effects of Nd_2_O_3_. In consideration of the wide application of Nd_2_O_3_ and its health risks to various systems, there is thus an urgent need to uncover the testicular toxicity and expression changes of key genes in the testicular tissue of mice exposed to Nd_2_O_3_. More researchers have analyzed specific samples exposed to toxins by sequencing techniques to identify key genes and regulatory pathways that cause diseases. LncRNAs that were greatly upregulated or inducibly expressed, such as lnc32058, lnc09522, and lnc98497, have been found in infertile male populations through RNA-seq [3,11]. Some lncRNAs, circRNAs, and mRNAs have potential value as candidate markers for the reproductive toxicity in male mice induced by tralitaxel [12]. Based on differential expression levels among mRNA, miRNA, lncRNA, and circRNA, some studies have proposed endogenous RNA competitive mechanisms (ceRNA) to form an interaction network to explore the regulatory mechanism of spermatogenesis [3,13,14]. These differentially expressed genes obtained through RNA-seq analysis will be especially important candidates for exploring the mechanism of male infertility. Therefore, we conducted RNA-seq analysis for mirRNAs/lncRNAs/circRNAs/mRNAs on the testicular tissue of mice exposed to Nd_2_O_3_ to find the key genes and regulatory pathways of male reproductive damage and explore the early biomarkers and mechanisms of reproductive damage caused by Nd_2_O_3_.

## 2. Materials and Methods

### 2.1. Animals and Sample Collection

Six-week-old C57BL/6 mice were obtained from China Speford (Beijing, China) Biotechnology Co. The experiment was approved by the Ethical Committee of the council of Baotou Medical College, Biomedical Lun Audit No. (009). Mice were kept under the same conditions (drinking water, food, environment) and were randomly divided into control and Nd_2_O_3_ dust groups, with 12 mice in each group. After one week of acclimatization, mice were anesthetized with tribromoethanol, and a single drip of NaCl (0.1 mg/mL) or Nd_2_O_3_ suspension (250 mg/mL) was administered using a non-exposed tracheal instillation method. Mice were executed by cervical dislocation after 35 days. Testis and epididymis were rapidly collected and removed for subsequent studies.

### 2.2. H&E Staining

Testes were immediately fixed in 4% paraformaldehyde solution, paraffin-embedded and sectioned, and stained with hematoxylin-eosin (H&E). Histological changes in testes were observed with a light microscope (Olympus, Tokyo, Japan).

### 2.3. Sperm Count and Morphology Assessment

We placed the mouse epididymis in 9 mL PBS at 37 °C constant temperature, clipped and incubated. Sperms were counted on a hemocytometer, and the sperm count was converted to the number of sperm per gram of epididymis (number of sperm per gram of epididymis = (n/4 × 10^4^ × 9)/weight of epididymis on both sides). Next, the sperm suspension was fixed with methanol for 5 min and then stained with 1% eosin for 1 h to produce a sperm smear. One thousand spermatozoa were observed in each sperm smear under light microscope, and fat-headed, double-headed, double-tailed, coiled-tailed, hookless, and folded-tailed types were used as the basis for determining sperm malformation. The number of sperm malformations was recorded to calculate the sperm malformation rate (sperm malformation rate = a number of malformed sperm/1000 × 100%).

### 2.4. RNA Library Construction and Sequencing

We isolated and purified total RNA from Nd_2_O_3_-exposed mice testicular tissue using Trizol reagent (Invitrogen, Carlsbad, CA, USA) according to the manufacturer’s procedure. RNA amount and purity of the sample were quantified using NanoDrop ND-1000 (NanoDrop, Wilmington, DE, USA). The RNA integrity was assessed by an Agilent Bioanalyzer 2100 (Agilent Technologies, Santa Clara, CA, USA). For miRNAs, we used TruSeq Small RNA Sample Prep Kits (Illumina, San Diego, CA, USA) to construct sequencing libraries. They were sequenced using Illumina Hiseq2000/2500 with a single-end 1 × 50 bp read length. For mRNA, lncRNA, and circRNA, we used rRNA depletion to construct a strand-specific library, sequenced on an Illumina NovaseqTM6000 (LC Bio, Hangzhou, China) with a single-end 2 × 150 bp read length. Construction and sequencing of RNA-seq libraries were performed by the Lianchuan Biological Company.

### 2.5. RNA Identification and Expression Analysis

We used Hisat (v 2.0.4) and ACGT101-miR software (LC Sciences, Houston, TX, USA) for quality control, genome matching, screening, and identification of raw data after the sequencing was completed. Differentially expressed genes were screened using a threshold of |log2(Foldchange)| ≥ 1 (two-fold difference multiplicity) and *p*-value ≤ 0.05, and the specific methods are described in the Appendix A. The expression of DEGs was clustered using log10(FPKM+1) to show their expression patterns and differences among groups of samples. We performed biological significance enrichment analysis of DEGs, including Gene Ontology (GO) and Kyoto Encyclopedia of Genes and Genomes (KEGG) analysis. A *p*-value ≤ 0.05 was considered as significant enrichment, and then the enrichment results were presented using ggplot scatter plots.

### 2.6. ceRNA Networks Analysis

We focused on analyzing miRNA sponge, using miRanda (v 3.3a) and TargetScan (v 3.3a) software (Whitehead Institute for Biomedical Research, Cambridge, MA, USA) for target gene prediction of DEGs. In the next step, lncRNA-miRNA-mRNA and circRNA-miRNA-mRNA ceRNA networks were constructed for differentially expressed lncRNAs, circRNAs, miRNAs, and mRNAs, and regulatory relationships were visualized with Cytoscape (v 3.0.1).

### 2.7. Protein–Protein Interaction (PPI) Network Construction

We performed protein–protein interaction networks (PPI) analysis of the differentially expressed mRNAs using the STRING (string-db.org) database. The PPI network was visualized with Cytoscape.

### 2.8. qPCR Confirmation

cDNA synthesis for mRNAs, lncRNAs, and circRNAs was performed using Prime Script RT kit (Vazyme Biotech, Nanjing, China). The expression level of the GAPDH was used as a control. For reverse transcription of miRNA, cDNA synthesis was performed using the Reverse Transcription System with U6 as an internal control. Relative gene expression was calculated by the 2^−ΔΔCT^ method. Samples were tested at least three times to normalize the expression levels of the genes of interest with good reproducibility. All primer sequences of mRNAs, miRNAs, circRNAs, and lncRNAs used for qPCR are listed in Appendix A. These were designed and synthesized by Biotech (Sangon Shanghai, China).

### 2.9. Statistical Analysis

All data are expressed as the mean ± SD. All statistical analyses were performed by *t*-test with GraphPad Prism 8GraphPad Software, Boston, CA, USA). Fold change values and *p*-values were used for assessing differences in gene expression. A *p*-value < 0.05 was considered statistically significant, and *p*-values < 0.01 were considered extremely statistically significant.

## 3. Results

### 3.1. Effects of Nd_2_O_3_ Exposure on Testis and Sperm Count and Sperm Morphology

To confirm the reproductive toxicity of Nd_2_O_3_ in vivo, we investigated the histological and sperm changes in the testis after Nd_2_O_3_ exposure. Compared with the control group, testicular tissues in the Nd_2_O_3_ intervention group exhibited reduced spermatozoa in the tubular lumen, marked abscission of spermatogenic epithelial cells, vacuolation of the epithelium, and abnormal Sertoli cells (Figure 1A,B). At the same time, the sperm count was significantly lower (Figure 1C), and the sperm morphology was abnormal. For example, we found folded-tail types, hookless types, and head and tail malformation (Figure 1D,E).

### 3.2. Expression Profiles of mRNA, miRNA, lncRNA, and circRNA

Using log2(Fold Change) ≥ 1 and *p* ≤ 0.05 as criteria for differential gene screening, we obtained 90 differentially expressed mRNAs, of which 30 mRNAs were upregulated (Figure 2A). There were 12 differentially expressed miRNAs and 10 downregulated miRNAs, as well as 82 differentially expressed circRNAs and 47 downregulated circRNAs (Figure 2B,C). There were 186 differentially expressed lncRNAs, of which 46 were known lncRNAs, and 140 were newly discovered lncRNAs (Figure 2D). In addition, differentially expressed lncRNAs were classified according to their different locations on chromosomes (Figure 2E). We performed a clustering analysis of the DEGs (Figure 2F–I), and the hierarchical clustering indicates that the closer the genes are, the more likely they are to have similar expression patterns and to be functionally related. We speculate that these differentially expressed genes may be biomarkers of Nd_2_O_3_-induced male reproductive toxicity, and they deserve further in-depth exploration.

### 3.3. GO and KEGG Analysis of DEGs

To reveal the functions of differentially expressed mRNAs, miRNAs, lncRNAs, and circRNAs, we performed an analysis of GO terms. The larger the rich factor, the higher the GO enrichment level. Mapping of the enriched significance (*p*-value) of the top 20 Go_terms revealed that many DEGs are enriched in pathways associated with testicular damage. mRNA is mainly involved in the following biological processes: the “lipid metabolic process”, “response to bacterium”, and “lyase activity” (Figure 3A). miRNAs are mainly involved in “protein binging”, the “G protein-coupled receptor signaling pathway”, and “membrane” activity (Figure 3B). Differentially expressed lncRNAs are mostly involved in “keratin filament” activity, “cellular zinc ion homeostasis”, and the “cellular response to zinc ion” (Figure 3C). GO annotations of the corresponding linear transcripts of circRNAs showed “spermatogenesis”, “chromatin binding”, and “positive regulation of GTPase activity” (Figure 3D).

KEGG focuses on biochemical pathways. We compared the DEGs in the KEGG database to identify signaling pathways in which the genes may be involved. mRNAs are involved in “Herpes simplex virus 1 infection”, the “PPAR signaling pathway”, and “Long-term potentiation” (Figure 3E). miRNAs are mainly involved in the “Wnt signaling pathway”, the “Phosphatidylinositol signaling system”, and the “FoxO signaling pathway” (Figure 3F). KEGG showed that lncRNAs are enriched in “Apoptosis-multiple species”, “Endocytosis”, and “Renin secretion” (Figure 3G). Pathway enrichment analysis of the host gene for circRNA focused on the “Jak−STAT signaling pathway”, the “Cell cycle”, and the “Phosphatidylinositol signaling system” (Figure 3H).

### 3.4. ceRNA Network Construction

We predicted the target binding relationships of miRNAs with mRNA, lncRNA, and circRNA and then constructed ceRNA networks based on common miRNA binding (Figure 4A). The figure shows the outward-to-inward lncRNA-miRNA-mRNA and circRNA-miRNA-mRNA networks. We constructed 412 such networks, with miR-7230-5p, miR-128-1-5p, and miR-29b-1-5p targeting the most genes.

### 3.5. PPI Analysis of Differential mRNAs

Our data analysis of differentially expressed mRNAs on the STRING database shows 88 nodes with 44 edges. A network plot of the genes analyzed for interactions using Cytoscape’s Betweenness (Figure 4B) consists of a total of 26 nodes, where larger nodes indicate the interaction of more mRNAs, and red nodes are the top 7 mRNAs. Secreted phosphoprotein 1 (Spp1), solute carrier family 34 (Slc34a1), lipoprotein lipase (Lpl), uromodulin (Umod), and cytochrome p450, family 2, subfamily e, polypeptide 1 (Cyp2e1) scored in the top five in the PPI network analysis.

### 3.6. Validation of Profiles via qRT-PCR

We performed partial validation of differentially expressed mRNAs, miRNAs, circRNAs, and lncRNAs. Due to the special loop formation mechanism of circRNAs, we selected circRNAs with a high number of back splice reads for validation. The qPCR results were consistent with the sequencing data (Figure 5A–D).

## 4. Discussion

Rare earth elements include 17 elements with similar physicochemical characteristics. Current research has demonstrated that rare earth elements have a variety of toxic effects, such as involvement in neurological disorders, oxidative stress, pneumoconiosis, kidney damage, and male infertility [15]. However, studies on the mechanisms of male infertility are still not fully described, with La and Ce being the most studied elements at present and Nd elements being less studied. It has been shown that cerium oxide nanoparticles (CeO_2_ NPs) increase the amount of Ce in mouse testes, causing pathological damage to testicular tissue and reduced sperm viability and survival [5]. Lanthanum oxide (La_2_O_3_) nanoparticles (NPs) are distributed in mouse testes and cross the blood–testis barrier, causing apoptosis of mouse spermatogenic cells [6,16]. Nd_2_O_3_ enters the body mainly in the form of dust or aerosols through the respiratory tract and then travels through the bloodstream to cause damage to various organs, of which the male reproductive organs are more sensitive than other organs. In the present study, we found that Nd_2_O_3_ caused damage to spermatogenic tubules, as well as epithelial vacuolization, germ cell abscission, reduced sperm counts, and spermatozoa exhibiting malformations in mice. These results indicated that Nd_2_O_3_ caused a decrease in sperm quality and reproductive toxicity.

The epigenetic changes that are involved in sperm degeneration are involved in the pathogenesis of reproductive toxicity. Functional non-coding RNAs (ncRNAs) have been identified as potential biomarkers for reproductive injury [3,11]. Based on the expression profile of non-coding RNA (ncRNA) in the serum of rare earth pneumoconiosis patients, our previous studies have demonstrated that the functions of some ncRNAs that were greatly upregulated or inducibly expressed were concentrated in the reproductive system [3]. The role and molecular mechanisms for ncRNA changes involved in reproductive injury after exposure to Nd_2_O_3_ need to be clarified. Thus, we performed RNA-seq analysis on mouse testicular tissues. As a result, 90 differentially expressed mRNAs, 12 miRNAs, 82 differentially expressed circRNAs, and 186 differentially expressed lncRNAs were identified, many of which are associated with male reproduction and spermatogenesis. We validated the partial sequencing results by RT-PCR, which was consistent with the results in the study, demonstrating the reliability of the results. RT-PCR results showed that four lncRNAs were greatly upregulated, and three lncRNAs were inducibly expressed in testicular tissue of mice exposed to Nd_2_O_3_. Most of these were related to reproductive function. Further, LncRNA NEAT1, a key parabolic component, is present in mouse mesenchymal cells and the punctate nuclei of GC-1 and GC-2 cell lines [17]. RT-PCR results showed miRNA-27b-3p, miRNA-19b-1, miRNA-128-1, and miRNA-446b-5p that were greatly upregulated or inducibly expressed by RT-PCR in the testicular tissue of mice exposed to Nd_2_O_3_. These are highly correlated with cell differentiation, proliferation, and apoptosis. These results show that overexpressed or downregulated RNAs participate in spermatogenesis, development, and maturation. miR-27b-3p is significantly downregulated in high glucose (HG)-treated GC-1 spg cells and improved autophagy and apoptosis in germ cells via the Gfp1/HBP pathway [18]. Furthermore, for these differentially expressed genes in the GO and KEGG databases, we further predicted the functions they perform, where circRNAs can perform similar functions as their homologs. Our results suggest that these genes are involved in the biological processes of spermatogenesis, the lipid metabolic process, and protein binding, and the pathways are enriched in signaling pathways such as Wnt, Jak-STAT, PPAR, and MAPK, all of which are important regulatory molecules in the process of spermatogenesis and blood–testis barrier remodeling [19,20,21]. These ceRNAs that were greatly upregulated or inducibly expressed provide valuable evidence for us to identify biomarkers and the mechanism of reproductive toxicity caused by Nd_2_O_3_.

The stability of the blood–testis barrier (BTB), one of the strongest blood–tissue barriers in the body, prevents harmful substances from entering the varicocele and is therefore important for the maintenance of normal male reproductive function. The BTB is composed of multiple junction types, including tight junction (TJ). In this study, we found through GO and KEGG analysis that some mirRNAs/lncRNAs/circRNAs were greatly upregulated or inducibly expressed, and gene pathways were enriched in the TJ, including actin cytoskeleton (actin tight junctions). These results indicate that the some mirRNAs/lncRNAs/circRNAs and the regulatory networks changed by Nd_2_O_3_ in the testicular tissue of mice may induce reproductive toxicity by affecting the tight linkage of the blood–testis barrier.

Spermatogenesis, as a process requiring precise regulation, is formed by spermatogonia undergoing mitosis and meiosis. The spermatozoa and testicular tissues of mice exposed to Nd_2_O_3_ showed significant spermatogonia shedding. Some studies have shown that there is some association between apoptosis and autophagy, which together influence spermatogenesis. Di-2-ethylhexylphthalate (DEHP)-induced autophagy may have a toxic effect on apoptosis in mouse GC-1 cells [22]. It has been reported that zinc oxide nanoparticles (ZnO NPs) may induce apoptosis and autophagy in mouse testicular mesenchymal cells through the activation of oxidative stress, and autophagy may play a protective role in this process [23]. In this study, ceRNA pathway analysis found that the ceRNA pathways caused by Nd_2_O_3_ were significantly enriched in apoptosis, cell cycle, and autophagy pathways. It is reasonable to assume that ceRNA is an important regulatory mechanism in germ cell division and differentiation. In addition, to better understand the relationship between mRNAs, we constructed a PPI network map using the STRING database, in which SPP1 scored highest, and it could interact with TLR2, which is expressed in the supporting cells of azoospermic males. Meanwhile, it activates the Myd88/NF-κB pathway with a protective effect on macrophage scorching [24,25]. Taken together, it is hypothesized that Nd_2_O_3_-induced reproductive toxicity is the result of a combination of multiple mechanisms and multiple physiological modulations, and ncRNAs and ceRNAs play important roles.

A variety of germ cells exist in the spermatogenic tubules of the mouse testis, and it has been shown that genes are specifically expressed in mouse spermatogonia, spermatocytes, spermatocytes, and supporting cells, regulating the process of spermatogenesis [26]. Therefore, we extracted mouse testis tissue for transcriptomic analysis based on traditional second-generation sequencing technology and obtained the average data of the transcriptome of all cells in the tissue. This tends to ignore cell-specific information and to obtain true cytogenetic information. The transcriptome analysis is now carried out at the single-cell level. Single-cell RNA sequencing (scRNA-seq) was performed on various types of germ cells at the spermatogenic stage as a way to assess intercellular variability and to reveal some bridge genes in spermatogenesis, such as DNA JC5B, BST1, and PTMs, which have been shown to play a role in male infertility [27]. Future studies combining scRNA-seq with RNA-seq analysis may provide new insights into integer cell heterogeneity and contribute to a comprehensive understanding of male reproductive processes as well as regulatory mechanisms.

## 5. Conclusions

In conclusion, our study obtained, for the first time, lncRNA/circRNA/miRNA/mRNA expression profiles in Nd_2_O_3_-exposed male mice and constructed ceRNA networks, such as lncRNA 4930509E16Rik-miR-19b-1-Cep72 and lncRNA Gm49793-miR-128-1. It provides a firm theoretical basis and a preliminary database for further studies on the biomarkers and molecular mechanisms regulating the reproductive toxicity exerted by Nd_2_O_3_.

## Figures and Tables

**Figure 1 toxics-11-00952-f001:**
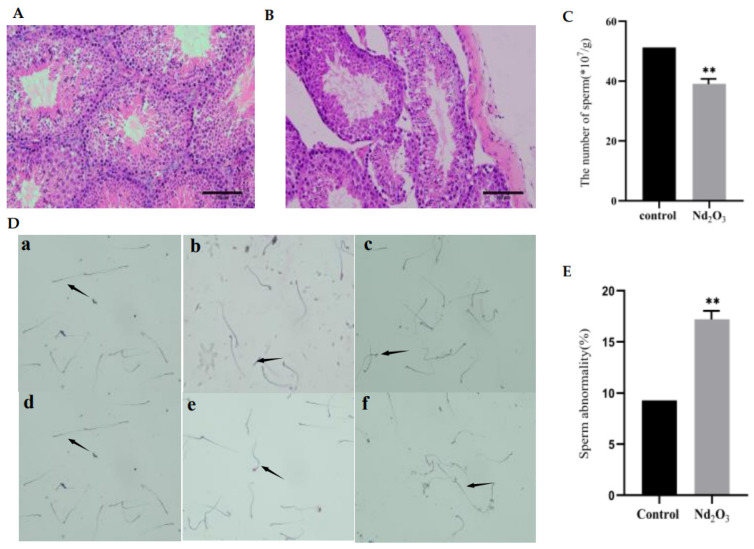
Effect of Nd_2_O_3_ on testis and sperm count and sperm morphology. C57BL/6 mice were given a dose of 0.1 mg/mL saline(control) or 250 mg/mL Nd_2_O_3_ using a non-exposed disposable tracheal injection lung irrigation method. The mice were cervically dislocated and executed after 35 days, and the testes and epididymis were taken. (**A**,**B**) Typical images of testicular histopathology by H&E staining (scale bar = 100 μm, *n* = 6). (**C**) Sperm count with cell counting plate (*n* = 12). (**D**) Representative images of sperm morphology by eosin staining, the arrows in the figure indicate the type of spermatozoa (scale bar = 50 μm, *n* = 12): (**a**), normal sperm; (**b**), folded-tailed sperm; (**c**), uncinate sperm; (**d**), fat-headed sperm; (**e**), double-tailed sperm: (**f**), double-headed sperm. (**E**) Statistical analysis of sperm deformities in mice (*n* = 12). All data were presented as mean ± SD. ** *p* < 0.01 compared with the control.

**Figure 2 toxics-11-00952-f002:**
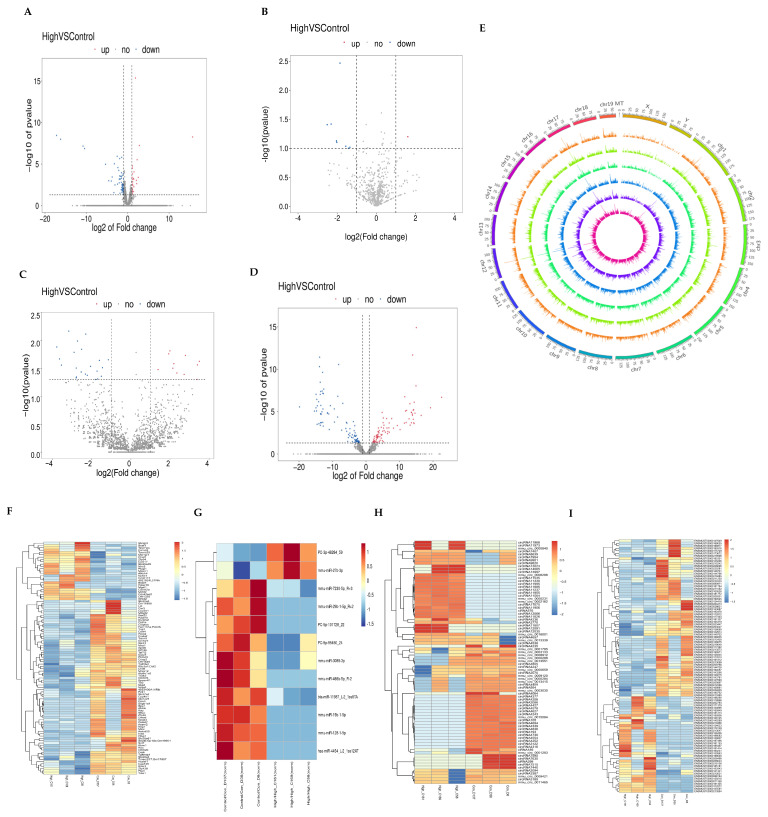
Analysis of differentially expressed mRNAs, miRNAs, circRNAs, and lncRNAs in mouse testis tissue by RNA-sequencing. (**A**) Differentially expressed mRNA volcano map. (**B**) Differentially expressed miRNA volcano map. (**C**) Differentially expressed circRNA volcano map. (**D**) Differentially expressed lncRNA volcano map. (**E**) lncRNAs were distributed by location in human chromosomes. (**F**) Differential expression of the mRNA heat map. (**G**) Differential expression of the miRNA heat map. (**H**) Differential expression of the circRNA heat map. (**I**) Differential expression of the lncRNA heat map.

**Figure 3 toxics-11-00952-f003:**
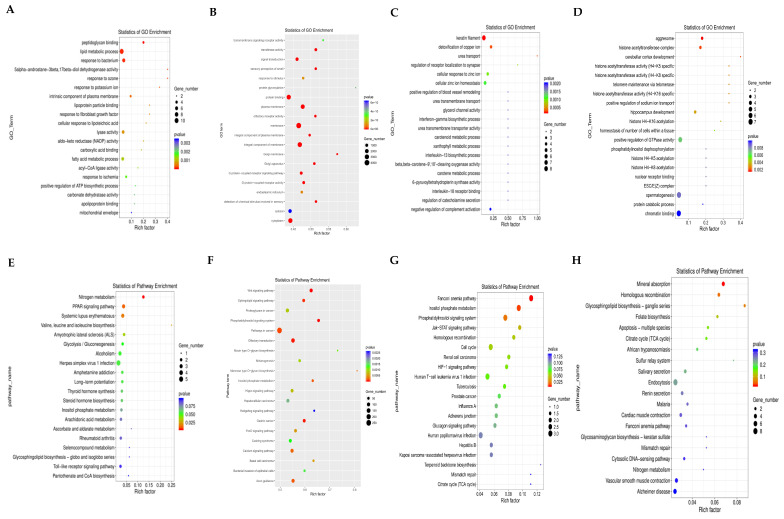
Differentially expressed mRNA, miRNA, circRNA, and lncRNA GO enrichment scatter plots and KEGG enrichment scatter plots in mouse testis tissue. (**A**) Differentially expressed mRNA GO enrichment. (**B**) Differentially expressed miRNA GO enrichment. (**C**) Differentially expressed circRNA GO enrichment. (**D**) Differentially expressed lncRNA GO enrichment. (**E**) Differentially expressed mRNA KEGG enrichment. (**F**) Differentially expressed miRNA KEGG enrichment. (**G**) Differentially expressed circRNA KEGG enrichment. (**H**) Differentially expressed lncRNA KEGG enrichment.

**Figure 4 toxics-11-00952-f004:**
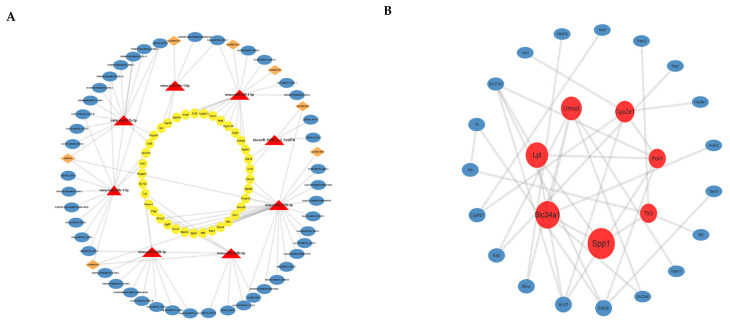
Analysis of ceRNA and PPI of differentially expressed genes. (**A**) The ceRNA regulatory network of lncRNA-miRNA-mRNA and circRNA-miRNA-mRNA. Blue circles represent lncRNAs, orange prisms represent circRNAs, red triangles represent miRNAs, and yellow circles represent mRNAs. (**B**) PPI analysis of differentially expressed mRNAs: the larger the circle, the more interacting genes, and the top seven mRNAs are in red.

**Figure 5 toxics-11-00952-f005:**
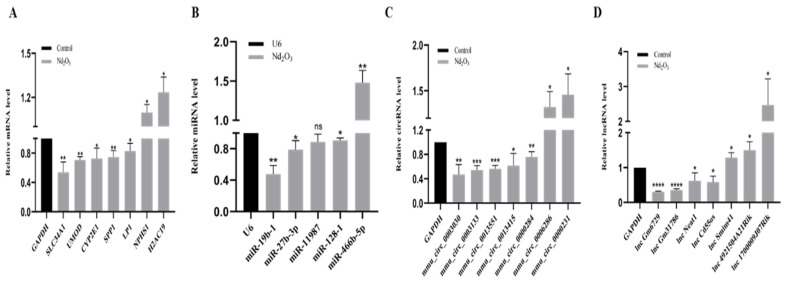
Validation of mRNA, miRNA, circRNA, and lncRNA expression by qRT-PCR. (**A**) mRNA qRT-PCR results. (**B**) miRNA qRT-PCR results. (**C**) circRNA qRT-PCR results. (**D**) lncRNA qRT-PCR results. * *p* ≤ 0.05, ** *p* ≤ 0.01, *** *p* ≤ 0.001, **** *p* ≤ 0.0001, ns: no significance.

## Data Availability

Data are available upon request.

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
