# Peer review of "RNA-Seq Analysis of Testes from Mice Exposed to Neodymium Oxide"

_toxics, 2023, doi:10.3390/toxics11120952_

Round 1
Reviewer 1 Report
Comments and Suggestions for Authors
1. The abstract of this article is inappropriate for the research article. It must be structured. It should start with an unambiguous aim, follow the list of materials and methods, then present the key results, and finally conclude with a conclusion. As stated in the manuscript, abstract sounds more like an introduction, and this must be corrected.
2. Introduction.
This section usually serves to introduce the readers to the subject of the investigation.
- Background of the major subject (what is known about the general subject of the study (about disease, method, biomarker, what is unknown or unsolved about the major subject of the study))
- A short overview of literature data
- Hypothesis: Declare what you hypothesized before you started with the investigation.
- Aim of the study: What was the aim of your study?
The aim should be the same as the one in the introduction. With the aim, you finalize the introduction. The authors must avoid presenting any data from the study here, including methodology, patients, or specific outcomes and expectations.
I do suppose that the statement “Therefore, to explore the key genes and regulatory pathways of male reproductive damage caused by 79 Nd2O3, we conducted RNA-seq analysis on testicular tissue of mice exposed to Nd2O3, target gene prediction, and GO and KEGG analysis on DEGs.”
- Each abbreviation must be expanded for the first time appearing in the text.
- Please clarify the aim as follows: The aim was to investigate the effect of rat testicular tissue exposure to Nd3O3 on the expression of the RNAs. ... (list out what you investigated: RNA panels, whole genom, etc.)
- METHODS
The statement of the Institutional Review Board about ethical issues related to the investigation on animals is missing at all.
More data about sample collection and storage is required for this type of study.
Almost all methods are described very superficially, with too little data for any scientist to repeat similar experiments.
Statistical analysis must be described in more detail.
4.RESULTS
Better resolution of figures is mandatory (i.e., text in figure 3 is nonreadable).
Comments on the Quality of English LanguageEnglish check is mandatory. Usage of article, commas and semi-commas is sometimes wrong.
Reviewer 2 Report
Comments and Suggestions for Authors
Title is informative and appropriate.
Abstract is well written.
Line 110: Please explain what the tissue sample was for RNA isolation.
About all software used in this article it is necessary to mention version and company and city.
In figures 1 E and 1 C: Why is not any error bars on control group?
Line 166: instead of magnification is better to identify size of scale bar.
Figure 1D is not clear.
Line 188 (E): The sentence is somewhat unclear.
Figure 2E is not mentioned and explained in result text.
Figures 2F-I are not clear to read.
Figures 3 and are not clear to read.
Figure 5B: What is U6?
It is better to start the discussion section with golden finding of this study and compare with previous studies and conclude each findings in a separate paragraph and final mini conclusion.
References are not written according to MDPI format.
Comments on the Quality of English Language
In some places, there are spelling errors and typographical errors, such as: lines 145, 203, 211, 247-249, 335, 346
Punctuations are different and some of them are English and some are Chinese fonts.
Round 2
Reviewer 1 Report
Comments and Suggestions for Authors
Response to the comments of Reviewer #1:
Comment No.1: 1. The abstract of this article is inappropriate for the research article. It
must be structured. It should start with an unambiguous aim, follow the list of materials and
methods, then present the key results, and finally conclude with a conclusion. As stated in
the manuscript, abstract sounds more like an introduction, and this must be corrected.
Response:In response to your suggestion, we have re-described the summary section and
marked it in red
(NOT SATISFIED, authors only add one phrase without substantial rewriting the abstract)
Reviewer 2 Report
Comments and Suggestions for Authors
I dont have more comments.
Author Response
thank you!
Round 3
Reviewer 1 Report
Comments and Suggestions for Authors
Improved as requested